# Risk Factors for Recurrence of Borderline Ovarian Tumours after Conservative Surgery and Impact on Fertility: A Multicentre Study by the Francogyn Group

**DOI:** 10.3390/jcm11133645

**Published:** 2022-06-23

**Authors:** Adele Ozenne, Marion De Berti, Gilles Body, Xavier Carcopino, Olivier Graesslin, Yohan Kerbage, Cherif Akladios, Cyrille Huchon, Alexandre Bricou, Camille Mimoun, Emilie Raimond, Lobna Ouldamer

**Affiliations:** 1Department of Gynecology, Tours University Hospital, 37044 Tours, France; ozenne.adele@gmail.com (A.O.); m.deberti@chu-tours.fr (M.D.B.); gilles.body@univ-tours.fr (G.B.); 2INSERM U1069, Université François-Rabelais, 37044 Tours, France; 3Department of Obstetrics and Gynecology, Hôpital Nord, APHM, Aix-Marseille University (AMU), Univ Avignon, CNRS, IRD, IMBE UMR 7263, 13397 Marseille, France; xavier.carcopino@icloud.com; 4Department of Obstetrics and Gynaecology, Institute Alix de Champagne University Hospital, Reims Champagne Ardennes University, 51100 Reims, France; ograesslin@chu-reims.fr (O.G.); eraimond@chu-reims.fr (E.R.); 5Department of Gynecologic Surgery, Jeanne de Flandre Hospital, CHRU LILLE, Rue Eugene Avinée, 59037 Lille, France; yohan.kerbage@chu-lille.fr; 6Department of Surgical Gynecology, Strasbourg University Hospital, 67000 Strasbourg, France; cherif.akladios@gmail.com; 7Department of Gynecology, CHI Poissy-St-Germain, EA 7285 Risques Cliniques et Sécurité en Santé des Femmes, Université Versailles-Saint-Quentin en Yvelines, 78000 Versailles, France; cyrillehuchon@yahoo.fr; 8Department of Gynecology, Bobigny University, AP-HP, Jean-Verdier Hospital, 93140 Bondy, France; alex.bricou@gmail.com; 9Department of Gynecology and Obstetrics, Lariboisiere Hospital, 750019 Paris, France; camille.mimoun@aphp.fr

**Keywords:** ovarian tumours, recurrence, conservative treatment, peritoneal staging, fertility

## Abstract

Introduction: Borderline ovarian tumours (BOT) represent 10–20% of epithelial tumours of the ovary. Although their prognosis is excellent, the recurrence rate can be as high as 30%, and recurrence in the infiltrative form accounts for 3% to 5% of recurrences. Affecting, in one third of cases, women of childbearing age, the surgical strategy with ovarian conservation is now recommended despite a significant risk of recurrence. Few studies have focused exclusively on patients who have received ovarian conservative treatment in an attempt to identify factors predictive of recurrence and the impact on fertility. The objective of this study was to identify the risk factors for recurrence of BOT after conservative treatment and the impact on fertility. Material and methods: This was a retrospective, multicentre study of women who received conservative surgery for BOT between February 1997 and September 2020. We divided the patients into two groups, the “R group” with recurrence and the “NR group” without recurrence. Results: Of 175 patients included, 35 had a recurrence (R group, 20%) and 140 had no recurrence (NR group, 80%). With a mean follow-up of 30 months (IQ 8–62.5), the overall recurrence rate was 20%. Recurrence was BOT in 17.7% (31/175) and invasive in 2.3% (4/175). The mean time to recurrence was 29.5 months (IQ 16.5–52.5). Initial complete peritoneal staging (ICPS) was performed in 42.5% of patients (*n* = 75). In multivariate analysis, age at diagnosis, nulliparity, advanced FIGO stage, the presence of peritoneal implants, and the presence of a micropapillary component for serous tumours were factors influencing the occurrence of recurrence. The post-surgery fertility rate was 67%. Conclusion: This multicentre study is to date one of the largest studies analysing the risk factors for recurrence of BOT after conservative surgery. Five risk factors were found: age at diagnosis, nulliparity, advanced FIGO stage, the presence of implants, and a micropapillary component. Only 25% of the patients with recurrence underwent ICPS. These results reinforce the interest of initial peritoneal staging to avoid ignoring an advanced tumour stage.

## 1. Introduction

Borderline ovarian tumours (BOT) were defined in 2003 by the World Health Organization (WHO) as tumours with “low malignant potential”. These are ovarian tumours of epithelial origin with proliferation of the stratified epithelial lining layer, but without stromal invasion and with a borderline contingent strictly above 10% [1].

They represent 10 to 20% of non-benign tumours of the ovary [1]. These tumours are rare but their incidence is increasing, estimated in France at 4.8/10,000 per year [2]. In contrast to malignant ovarian tumours, the incidence of BOTs is increasing, probably due to improvements in histological diagnostic performance [3]. Among women with BOT, one third are under 40 years of age [4]. In today’s society, where age at first pregnancy is increasing [5], it is not uncommon for BOT to be diagnosed in a patient with unfulfilled motherhood desire [6].

The current challenge of surgical management of BOT is to allow ovarian conservative treatment while performing complete peritoneal staging in order to assess the risk of recurrence and to limit recurrence in an invasive form. It is currently accepted that, regardless of tumour stage, ovarian conservative surgery is recommended for women wishing to preserve fertility or endocrine function [7,8,9]. However, while overall survival does not seem to be negatively affected [8], the risk of local recurrence appears to be increased for those patients receiving conservative treatment [10,11].

The pregnancy and birth rate after conservative surgery is high, even after a second conservative surgery for local recurrence [12,13], though there is no consensus on the ideal surgical procedure to treat these patients with pregnancy desire, presenting a tumour recurrence after conservative surgery. Some authors have reported higher rates of recurrence in women with cystectomy, thus advocating unilateral salpingo-oophorectomy (USO), while others point out a possible impairment of fertility associated with oophorectomy [14].

It is now accepted that conservative treatment is a risk factor for recurrence of BOT [7,15,16,17,18,19,20,21,22,23,24,25,26,27,28,29,30,31,32,33,34,35]. Fertility-sparing surgery has become a major issue in young patients with BOT with desire for pregnancy. To date, many studies have attempted to identify predictive factors of recurrence of BOT regardless of surgical strategy, but few have focused exclusively on patients who have had ovarian conservative treatment [13,19,20], to identify these risk factors for recurrence and to compare the prognosis of these tumours and pregnancy outcomes: in 2019 and 2020, two observational studies conducted by Chevrot et al. [13] and Plett et al. [20], including respectively data from 52 and 95 patients, had as their main objectives analysing the risk factors of recurrence and the fertility of patients treated by ovarian conservative surgery.

These two studies, although relevant, involved limited numbers of patients and did not look at as many histological characteristics. These studies focused their analysis on FIGO stage, histological subtype, and type of conservative treatment. In clinical practice, it seems relevant to study the different histological features of BOT that may influence the occurrence of recurrence.

Thus, in this study, we performed a retrospective multicentre analysis of a population of patients at high risk of recurrence, with the aim of identifying the risk factors of recurrence of BOT in these patients with desire of pregnancy, treated by conservative surgery and to observe the impact of this surgery on fertility in order to improve the therapeutic management, and to adapt the surveillance.

The objectives of this study were to identify the risk factors for recurrence of BOT and patients at high risk of recurrence and to evaluate the impact on fertility.

## 2. Materials and Methods

### 2.1. Population

We propose a clinical study based on a retrospective multicentre database updated in eight French cancer centres belonging to the FRANCOGYN study group: the University Hospital Centres of Tours, Jeanne de Flandre of Lille, Strasbourg, Reims, Marseille, the Lariboisière Hospital, the Intercommunal Hospital Centre of Poissy, and the “Jean Verdier” Hospital.

From the overall population, we focused on patients who had undergone conservative surgery: patients included were had BOT, treated with ovarian conservative surgery between February 1997 and September 2020, with FIGO stage I to IV tumours, and older than 18 years of age. All cases were reviewed by an experienced pathologist and defined according to WHO criteria.

We excluded patients with a borderline contingent less than 10%, those who had undergone radical surgical treatment, and those for whom data on recurrence status and survival were not available.

All patients included in the study underwent a preoperative evaluation that included an interview, clinical examination, pelvic ultrasound imaging, and pelvic MRI in case of an undetermined ovarian lesion on ultrasound.

### 2.2. Definitions and Procedures

Epidemiological data of the patients were recorded. The early stages corresponded to FIGO stage I, and the advanced stage corresponded to FIGO stages II, III, and IV [7].

For the management, we collected data regarding surgical management: date and type of surgery, along with the approach and performance of complete peritoneal staging.

Conservative treatment was defined according to French guidelines [7] as preservation of the uterus and at least part of the ovary allowing preservation of fertility or endocrine function.

The recommended peritoneal staging included careful inspection of the peritoneum, peritoneal cytology, multiple peritoneal biopsies, omentectomy, and appendectomy in case of mucinous type tumour or macroscopically pathological appearance [35]. It was called initial complete peritoneal staging (ICPS). It was to be performed at the time of the first surgery if the diagnosis of BOT was confirmed peroperatively by frozen examination. However, in case of secondary histological diagnosis, the benefit of a new surgery for restaging was discussed if there was a risk of existence of extra-ovarian implants (serous tumour with micropapillary (MP) component) or incomplete abdominopelvic exploration for serous tumours, and in case of non-visualized appendix or initial cystectomy for mucinous tumours, allowing then to re-evaluate upwards the initial FIGO stage while taking into account the potential morbidity associated with this surgical revision. Otherwise, the peritoneal staging was considered “incomplete”.

Regarding histological data, for serous tumours, peritoneal implants were classified as non-invasive and invasive (IPI) [29,31,32,33,34,35,36,37,38,39,40]. Because the classification of gynaecological tumours was revised in 2014, in this work, we will maintain this nomenclature of peritoneal implants [21]. Invasive implant was defined by invasion and destruction of adipose or peritoneal tissue with desmoplastic stroma-reaction. The term implant should not be used in the context of mucinous-type BOT, since extraovarian disease associated with mucinous BOT should be considered a metastasis of ovarian or other organ origin [41].

We also noted the existence of a micro-papillary (MP) component, defined by its presence over an area of at least 5 mm on the maximum surface [33,34,41]. For serous tumours, the classical or micro-papillary histologic subtype was specified.

The term microinvasion referred to invasion of less than 5 mm, without desmoplastic reaction stroma, which can be seen in all morphological subtypes of BOTs [21]. We also noted the presence of lymphovascular invasion of the ovarian capsule. Tumour stage was defined according to the 2014 FIGO classification of ovarian tumours [22].

Desire for pregnancy, post-therapy fertility, number of pregnancies, as well as obstetrical outcome in case of pregnancy, were also collected.

### 2.3. Definitions and Classifications of Recurrence 

In agreement with the literature [42], we defined recurrence as a relapse of the disease without distinction between the occurrence of a new borderline tumour, extra-ovarian implants, or an infiltrating tumour.

In case of clinical suspicion of recurrence, patients received a clinical examination, measurement of tumour markers (CA125, CA19.9), and appropriate radiological examinations (i.e., pelvic ultrasound, CT scan, pelvic MRI, or other as indicated). Confirmation of the diagnosis of recurrence was by pathological analysis by biopsy or surgery.

The time to recurrence in months was defined as the time between the date of the first surgery and the date of discovery of the recurrence.

Recurrence-free survival (RFS) was defined as the time from the date of histological diagnosis to the date of first recurrence and was censored from the date of last news or death in the absence of recurrence. Overall survival (OS) was defined as the time from the date of histological diagnosis to the date of death from any cause.

Patients were divided into two groups according to whether they had recurrence (R group) or not (NR group).

### 2.4. Statistical Analysis 

Univariate analysis was performed to assess patient characteristics (age, BMI, history), tumour characteristics (bilateral involvement, MP contingent, FIGO stage, peritoneal implants), and surgical modalities (approach, type of surgery). To compare categorical variables, we used the chi2 test or Fisher’s exact test when the conditions for chi2 validity were not required. To compare a continuous variable with a categorical variable, we used Student’s test, or the Wilcoxon test when the variable did not follow a normal distribution.

Multivariate analysis was performed using a logistic regression model and including factors that were significant in the univariate analysis.

For the survival data, the curves were made by the Kaplan–Meier method. The survival analysis was performed with a Cox model allowing the calculation of Odds Ratio (OR), which was calculated with a 95% confidence interval (95% CI). The results of the tests were considered significant when the *p* value was less than 0.05.

Statistical analyses were performed using R.3.1.2 software (Hmisc package, Design and Survival libraries).

## 3. Results

### 3.1. Characteristics of the Study Population

From February 1997 to September 2020, 639 women were managed for BOT. Data from 175 patients meeting the inclusion criteria were analysed.

These data came from eight French centres, members of the FRANCOGYN group: the University Hospitals of Tours (*n* = 44), Lille (*n* = 40), Strasbourg (*n* = 28), Reims (*n* = 9), Marseille (*n* = 11), Lariboisière (*n* = 22), Poissy (*n* = 15), and “Jean Verdier” (*n* = 6).

Among these patients, 35 had a recurrence (group R, 20%) and 140 had no recurrence (group NR, 80%).

The epidemiological characteristics of the patients are summarized in Table 1.

The mean age of the patients at the time of diagnosis was 30 years (IQ 25.5–34), and in 93% of cases, the patients were under 45 years of age. In the cohort, 58.8% were nulliparous (*n* = 103). The average parity was near 1 child per woman and the average BMI was 23.02 kg/m^2^ (17.0–45.2).

A first surgery by laparoscopy was performed in 70.8% of cases (*n* = 124), a complete initial peritoneal staging was performed according to recommendations for 42.5% of the surgical procedures. The intraoperative rupture rate was 20.2% (*n* = 32).

The histological characteristics are summarized in Table 2.

The average patient follow-up was 30 months (IQ 8–62.5). The overall recurrence rate was 20% (*n* = 35). Recurrence was of the borderline type in 89% of cases (*n* = 31, or 17.7% of the total population) and of the invasive type in 11% of cases (*n* = 4, or 2.3% of the total population).

The mean time to overall recurrence was 29.5 months (IQ 16.5–52.5), the mean time to BOT recurrence was 53.7 months (IQ 17–60.5), and the mean time to invasive recurrence was 115 months (IQ 49.2–175).

### 3.2. Characteristics of the Study Population According to Recurrence Status

The epidemiological data of each group according to recurrence status are summarized in Table 3 and histological data in Table 4.

Patients with recurrence after conservative surgery were significantly younger than those without recurrence (28 years versus 31.5 years, *p* = 0.02). There was significantly more preoperative rupture in the recurrence group (6% versus 0% *p* = 0.049).

Significantly more ICPSs were performed in the non-recurrent group (NR group) compared to the recurrent group (R group) (46% versus 25%, *p* = 0.05).

The univariate and multivariate analysis of factors associated with recurrent is summarized in Table 5.

In univariate analysis, age less than 35 years (OR = 10.4 (95% CI 1.41–76.2(; *p* = 0.02), stage greater than or equal to a FIGO II stage (OR = 10.2 (95% CI 2.22–47.3); *p* = 0.002), the presence of an MP component for serous tumours (OR = 4.35 (95% CI 1.82–10.3); *p* = 0.0009) and the presence of implants (OR = 2.67 (95% CI 1.24–5.71); *p* = 0.01) appeared to significantly influence the risk of recurrence.

Parity appeared to be a protective factor for recurrence, each pregnancy prior to surgery seemed to significantly reduce the occurrence of recurrence by 36% (OR = 0.36 (95% CI 0.15–0.86); *p* = 0.02).

For mucinous tumours, there was no association between the presence of in situ carcinoma and the occurrence of recurrence (*p* = 0.27).

In multivariate analysis, age less than 35 years (OR = 1.40 (95% CI 1.22–159); *p* = 0.034), nulliparity (OR = 8.04 (95% CI 1.73–37.4); *p* = 0.007), a PM component (OR = 8.47 (95% CI 2.42–29.6); *p* = 0.0008), as well as the presence of peritoneal implants (OR = 5.52 (95% CI 1.8–17.0); *p* = 0.003) appeared to be factors significantly influencing the occurrence of recurrence (Table 5).

Of the 175 patients who underwent surgery, 45 had a desire for subsequent pregnancy. The post-surgery fertility rate was 67% in patients with a desire to become pregnant, with no significant difference between the two groups, and 51% of patients (*n* = 23) gave birth.

## 4. Discussion

In this study, we analysed data from 175 patients with conservative management for BOT, 35 of whom had a recurrence (20%).

Our study identified four risk factors for recurrence of BOT after conservative treatment in multivariate analysis: age at diagnosis, nulliparity, a micropapillary component, and the presence of peritoneal implants.

This series is one of the largest reported to date on the conservative treatment of BOT. The results of our study contain two important messages. The first concerns the oncological results and safety of conservative treatment of BOT, the second concerns the risk factors for recurrence of BOT.

To date, it is accepted that conservative surgery is associated with an increased risk of recurrence [7]. Therefore, we know that we are interested in a subpopulation belonging to a group at high risk of recurrence. Numerous studies have identified conservative surgery as a risk factor for recurrence of BOT, though without impact on overall patient survival [14,15,16,43,44]. A 2014 meta-analysis by Vasconcelos et al. [44], highlighted for serous tumours, that among the different conservative treatments, USO was associated with a significantly lower rate of recurrence than simple cystectomy, without impact on survival. For bilateral tumours, the recurrence risk was similar after bilateral cystectomy or USO combined with contralateral cystectomy, so the former option was recommended. For mucinous tumours, USO was recommended over simple cystectomy [45] with a significantly lower recurrence rate for USO.

These results were confirmed by two studies: one by Ouldamer et al., the other by Bendifallah et al. [17,42], proposing a predictive model of the risk of recurrence, in which conservative surgery and in particular cystectomy showed a strong association with an increased risk of recurrence at 5 years (respectively HR = 10.25 95% CI (5.2–20.22)) and HR = 11.35 95% CI (4.01–32.08)).

The recurrence rate was significant (20%) but fortunately, the vast majority of recurrences were of the borderline type (89% of recurrences), and 4 patients (11% of recurrences) had a recurrence of the invasive type. In a review of the literature by Morice et al. [31] regarding the risk of recurrence of the invasive form of BOT, among 1800 conservative surgeries performed on BOT, only 10 recurrences were reported for the early stage. It is difficult to determine whether such recurrences could be related to the natural history of the tumour or to the conservative approach.

The overall mean time to recurrence for BOTs was 29.5 months (IQ 16.5–52.5) and the 5-year and 10-year recurrence-free survival were 74.2% and 58%, respectively. In recent studies [13,16,20], the results are similar with a mean time to recurrence between (30–36 months) (Table 6) and a recurrence-free survival between (60–80%) at 5 years and between (42–74%) at 10 years. Our results support the interest of conservative surgery for young patients with BOT with a desire for pregnancy. Nevertheless, there is a risk of invasive recurrence and therefore a risk of death (0.6% in our cohort, identical to the literature, estimated at 0.5% after conservative surgery) [14]. During the first postoperative years, it seems difficult to determine the adequate modalities and duration of surveillance. Careful follow-up is therefore mandatory, and patients should be informed of this rare risk.

The second important point concerns the risk factors for recurrence: 

Patients who had recurred were significantly younger than patients who had not recurred, and an age below 35 years appeared to be a risk factor for recurrence of BOT in multivariate analysis. In the literature, the young age of patients is frequently found to be a prognostic factor for recurrence of BOT [42,46]. In multivariate analysis, it was the only factor found in the study by Uzan et al. [46]. As they have a longer life expectancy, young patients would theoretically have a higher risk of recurrence, since recurrence of BOT can occur after a long period, which may be more than 15 years after the initial management [47].

Nulliparity also appears to be a factor influencing the occurrence of recurrence of BOT. To our knowledge, we have not found any study that has analysed the influence of nulliparity on recurrence of BOT.

In our study, and as expected, an advanced FIGO stage was found to be a risk factor for recurrence of BOT after conservative surgery. This result is consistent with the literature, as many authors have shown that advanced FIGO stage (≥II) was a factor in recurrence risk and decreased recurrence-free survival after conservative treatment, even in the case of complete surgery [15,16,19,28,37,43,48,49].

The presence of implants was also a risk factor for recurrence of BOT in multivariate analysis. We did not find a difference according to whether the implants were invasive or not. In the literature, invasive peritoneal implants (IPI) are often found to be predictive factors of recurrence of BOT associated with a significantly decreased recurrence-free survival [50,51,52]. It is unclear whether these IPIs were suspected at the time of exploration of the peritoneal cavity or whether they were confirmed histologically by routine biopsies [7]. In the latter case, it is easy to understand the importance of peritoneal staging because the presence of IPIs raises the FIGO classification and may change the therapeutic management and surveillance.

Our study found a significantly higher rate of MP component in the R group (32% versus 12%, *p* = 0.04). The presence of an MP component appeared to be a factor associated with the risk of recurrence of BOT in multivariate analysis. There are few studies on the conservative management of MP serous BOTs, and opinions differ. For Vasconcelos et al. [47], the MP subtype, all FIGO stages included, was associated with more lethal recurrence than conventional advanced serous BOT (OR = 0.501; *p* = 0.003). Conversely, Uzan et al. [49], in a series of high-grade serous BOT, showed that the MP pattern was not associated with a poor prognosis (*p* = 0.8) and that the only factor predicting recurrence in the cohort was the use of conservative treatment (*p* = 0.007).

In our work, the type of conservative surgery did not appear to be a factor influencing the occurrence of recurrence in multivariate analysis. Our work seems to be in agreement with the literature. A recent Italian study, representing the largest observational study of BOT after conservative surgery to date, conducted by Delle Marchette et al. [19], showed no association between the type of conservative surgery and the risk of recurrence (HR = 1.34 (95% CI 0.98–1.81); *p* = 0.06). A French meta-analysis [14], however, and a large German multicentre series [26], showed that ultraconservative surgery increases the risk of recurrence (OR = 2.36 (95% CI 1.22–4.55); *p* = 0.002). Nevertheless, this does not mean that oophorectomy should be preferred to cystectomy, as the use of the latter procedure also increases the fertility rate. A recent phase III trial by Palomba et al. [53,54] (the only one concerning BOT to date), showed that bilateral cystectomy had better fertility outcomes compared to USO associated with contralateral cystectomy (OR = 8.05 (95% CI 1.20–9.66); *p* < 0.01), with a shorter surgery-to-pregnancy time (*p* < 0.02), despite a significantly shorter RFS (*p* < 0.001). Preservation of maximum healthy ovarian volume (and follicles) should therefore be proposed to improve fertility outcomes. Cystectomy therefore appears to be the preferred surgical procedure for young women wishing to preserve their fertility, whatever the FIGO stage, whereas oophorectomy is recommended for postmenopausal women [7]. There are no clear criteria to date to suggest that performing cystectomy for BOT is detrimental to long-term survival.

In our study, only 42.5% of patients underwent initial complete peritoneal staging (ICPS). Among them, an ICPS was performed in only 25% of patients with recurrence versus 46% of patients without recurrence. It should be noted that there was significantly less recurrence after SPCI. Our results are in agreement with the meta-analysis of Shim et al. [55] based on observational studies, which shows that IPCS appears to significantly reduce recurrence in patients with BOT (OR = 0.64 (95% CI 0.47–0.87); *p* < 0.05). Several studies have found an increase in recurrence rate proportional to the number of missing procedures, with omentectomy being the procedure with the greatest impact (HR = 1.91 (95% CI 1.15–3.19); *p* = 0.013) [55,56]. Similar to our results, these studies also found a low rate of ICPS in accordance with the recommendations (between 31.7 and 49.7%), which represented a bias in the analysis of recurrences with a probable underestimation of the FIGO stage and a risk of performing an incomplete resection with a tumour residue. The benefit of ICPS would be to not ignore the presence of peritoneal implants in order to not under-stratify the patients.

On the other hand, our study did not find any significant difference in the risk of recurrence after restaging (*p* = 0.11), which is also in line with the literature since the meta-analysis by Chevrot et al. [57] showed that restaging did not influence the risk of recurrence of BOT (OR = 0.88 (95% CI 0.41–1.92); *p* = 0.76). The latest recommendations on the benefits of a new surgical staging, however, are clear [7], even though this repeat procedure exposes patients to anaesthetic and surgical risks. It would appear from our study and the literature that this strategy has no impact on RR. The benefit of restaging is not clear but should be discussed in patients with serous BOTs with MP pattern or for whom visual exploration of the peritoneal cavity is incomplete [41,58].

The studies are mainly concerned with serous type BOT, Table 6 show that pregnancy rates are higher in the Asian series of Park et al. [59], Fang et al. [43], and Chanson et al. [60], in which the percentage of mucinous tumours treated conservatively is higher than in the other series: the pregnancy rates were 73% and 68%, respectively, and 80% and more than half of the patients who underwent conservative surgery were managed for mucinous type tumours. Unfortunately, to our knowledge, data regarding mucinous tumours and fertility after conservative surgery are scarce.

The prognosis of advanced stage BOT after conservative treatment is more reserved, even though this therapeutic option appears to be encouraged overall [20,48,61,62,63]. For Uzan et al. [62] and Gouy et al. [48], despite a high recurrence rate (respectively 56% and 58%), the spontaneous pregnancy rate was good after conservative treatment (35.9% and 68.9%), and overall survival was excellent (100% at 5 years and 92% at 10 years), even though SRH was significantly reduced compared to early stages (HR = 25 (95% CI 7.7–95); *p* < 0.001) for the second team. Darai et al. [14], reported a lower spontaneous pregnancy rate than early-stage tumours (34% vs. 54%) with a high local and lethal RR (38% (95% CI 26–50%)) for advanced stages compared to early stage (13% (95% CI 10–16%)). As shown in Table 6, although the local recurrence rate is high [6,64], the prognosis is good, and fertility seems to be preserved with a post-surgery pregnancy rate between 35 and 70% depending on the series [48,61] and no impact on overall survival is found. However, in view of the high risk of local and invasive recurrence, careful and close surveillance with a prolonged follow-up period seems necessary.

While the risk of recurrence is significantly associated with fertility-preserving surgery, this does not appear to have an impact on patient survival. Invasive recurrence, influencing patient survival, remains a rare event after conservative treatment. Most recurrent lesions are non-invasive in nature and can be easily treated with conservative surgery. Therefore, it appears that the increased recurrence rate after conservative surgery does not impact patient survival.

While peritoneal staging does not appear to have an impact on the recurrence of BOT, ICPS seems on the contrary to decrease the risk of recurrence. Our results reinforce the interest of ICPS in order not to ignore an advanced tumour stage, in particular, with the presence of peritoneal implants which represents a factor influencing the occurrence of recurrence, and thus, to limit the recurrence of BOT after conservative surgery. It would therefore seem desirable to refer patients to an expert surgical oncology centre in order to optimise and harmonise the overall management of these patients.

## Figures and Tables

**Table 1 jcm-11-03645-t001:** Clinical, biological and surgical characteristics of the population.

	*n* = 175
**Demographic data**	
**Mean Age at diagnosis** in years, median (IQ)	30 (25.5–34)
Age ≤ 35 years, *n* (%)	139 (79.4)
Age ≤ 45 years, *n* (%)	**165 (94.2)**
**Mean BMI** (kg/m^2^), median (IQ)	23.02 (21.0–27.26)
**Parity** median (IQ)	0.78 (0.0–2.0)
**Nulliparity**, *n* (%)	**103 (58.8)**
**Antecedent**, *n* (%)	
Unilateral ovariectomy	7 (4.0)
Familial history of breast cancer (NA = 15)	30 (18.7)
Familial history of ovarian cancer (NA = 15)	4 (2.5)
**Ca125** (UI/mL), median (IQ)	34.3 (15.9–125.0)
NA	50
**Ca 19-9** (UI/mL), median (IQ)	12.8 (5.1–33.0)
NA	9
**Ultrasound size of the ovary** (mm), median (IQ)	90 (60–150)
**Surgical data**	
**Surgical Route**	
Laparoscopy	**124 (70.8)**
Laparotomy	42 (24)
Laparo-conversion	8 (4.6)
NA	1 (0.5)
Preoperative Rupture	2 (1.1%)
Peroperative Rupture, *n* (%) (NA = 15 + 2 preoperative)	**32 (20.3)**
**Type of surgery**	
Unilateral Cystectomy	43 (24.6)
Unilateral Ovariectomy	109 (62.3)
Bilateral cystectomy	6 (3.4)
Cystectomy and contralateral ovariectomy	17 (9.7)
**Peritoneal staging**, *n* (%)	
Initial (ICPS)	**74 (42.3)**
Secondary	97 (55.4)
Incomplete	4 (2.2)

Data: means (minimum–maximum), and number (%), IQ: interquartile range, NA: missing data, mm: millimetre, BMI: body mass index, CEA: carcinoembryonic antigen, Ca 125: cancer antigen 125, Ca 19-9: carbohydrate antigen 19-9, CPS: initial complete peritoneal staging.

**Table 2 jcm-11-03645-t002:** Histological characteristics.

	*n* = 175
Histological characteristics	
**Histological type**, *n* (%)	
Serous	**80 (45.7)**
Mucinous	**89 (50.8)**
Endometrioïd	3 (1.7)
Sero-mucinous	2 (1.1)
**Bilateral lesions**	23 (13.1)
**FIGO stage**, *n* (%)	
IA	**121 (69.1)**
IB	**6 (3.4)**
IC	**26 (14.8)**
II	4 (2.3)
III	16 (9.1)
IV	0
NA	2 (1.1)
**Positive peritoneal cytology**	27 (15.4)
**Micro-papillary component for serous type (MP)**, *n* (%)	
Yes	18 (22.5)
**Implants**	
Presence of implants	25 (14.3)
Invasive peritoneal implants (IPI)	2 (1.1)
**Micro-invasion**	
Yes	12 (8.6)
NA	35 (20)
**Recurrence**	
BOT	31 (17.7)
Invasive	4 (2.3)
**Follow-up (months), median (IQ)**	**30 (8–62.5)**
**Time to recurrence (months), median (IQ)**	**29.5 (16.5–52.5)**

Data: means (minimum–maximum), and number (%), IQ: interquartile range, cm: centimetres, FIGO: International Federation of Gynaecology and Obstetrics, MP: micro-papillary, NA: missing data.

**Table 3 jcm-11-03645-t003:** Epidemiological and surgical characteristics of the population by recurrence status.

	R Group	NR Group	*p*-Value
	*n* = 35	*n* = 140	
**Demographic characteristics**			
**Age mean (years)**	28 (±4)	31.5 (±5)	**0.02**
**BMI mean (kg/m^2^)**	22.5 (20.9–26.05)	23.2 (21–27.6)	0.52
<25			
**Nulliparity**, *n* (%)	27 (77.1)	76 (54.3)	0.15
**Family history**			
Familial history of breast cancer	11	19	**0.03**
Familial history of ovarian cancer	0	4	0.49
**Ca125 (UI/mL)**	68 (38–177.5)	30 (15–91.9)	0.48
**Ca19.9 (UI/mL)**	21 (7.5–108.5)	12.6 (4.9–27.7)	0.97
**Operative data**			
**Surgical route**, *n* (%) NA = 1			
Laparoscopy	27 (77.1)	97 (69.3)	
Laparotomy	6 (17.1)	36 (25.7)	0.50
Laparoconversion	1 (2.8)	7 (5)	
Preoperative rupture	2 (5.7)	0 (0)	**0.049**
Peroperative rupture	8 (22.8)	24 (17.1)	0.69
**Type of surgery**, *n* (%)			
Unilateral cystectomy	6 (17.1)	37 (26.4)	
Unilateral salpingo oophorectomy	19 (54.3)	90 (64.3)	
			**0.02**
Bilateral Cystectomy	3 (8.6)	3 (2.1)	
Cystectomy and contralateral oophorectomy	7 (20)	10 (7.1)	
**Peritoneal staging**, *n* (%)			
Initial (ICPS)	9 (25.7)	65 (46.4)	**0.049**
Secondary	24 (68.6)	73 (52.1)	0.11

Data: means (minimum–maximum), and number (%), IQ (interquartile range), mm: millimetre, BMI: body mass index, CEA: carcinoembryonic antigen, Ca 125: cancer antigen 125, Ca 19-9: carbohydrate antigen 19-9, NA: missing data.

**Table 4 jcm-11-03645-t004:** Histological characteristics according to recurrence status.

	Group R	Group NR	*p*-Value
	*n* = 35	*n* = 140	
**Histological characteristics**			
**Histologic type**, *n* (%)			
Serous	21 (60)	59 (42.1)	
Mucinous	12 (34.3)	77 (55)	0.12
Endométrioïd	1 (2.8)	1 (0.7)	
Sero-mucinous	1 (2.8)	1 (0.7)	
**Bilateral lesion**	9 (25.7)	5 (3.6)	**0.02**
**FIGO stage**, *n* (%)			
IA	18 (51.4)	103 (73.6)	
IB	3 (8.6)	3 (2.1)	
IC	5 (14.3)	21 (15)	**0.01**
II	2 (5.7)	2 (1.4)	
III	7 (20)	9 (6.4)	
Early stage	26 (74.3)	127 (90.7)	**0.008**
Advanced stage	9 (25.7)	11 (7.8)	
NA = 2			
**Micro-papillary component (MP)**	8 (32)	10 (12)	**0.04**
**Implants**, *n* (%)			
Total	12 (34.3)	13(9.3)	**0.0004**
IPI	0 (0)	2 (1.4)	0.86
**Micro-invasion**, *n* (%)	3 (8.6)	9(6.4)	0.94

Data: means (minimum–maximum), and number (%); FIGO: International Federation of Obstetrics and Gynaecology; IPI: invasive peritoneal implants, NA: missing data.

**Table 5 jcm-11-03645-t005:** Risk factors for recurrence in univariate and multivariate analysis.

		OR (IC 95%)	*p*-Value
Univariate analysis			
Age at diagnosis ≤ 35 years	No	Reference	
	Yes	**10.4 (1.41–76.2)**	**0.02**
Nulliparity	No	Reference	
	Yes	4.51 (1.91–10.7)	**0.0006**
Parity		**0.36 (0.20–0.66)**	**0.0008**
Unilateral lesion	No	Reference	
	Yes	**0.33 (0.14–1.81)**	**0.01**
Ca125 (UI/mL)	≥35	Reference	
	<35	**0.17 (0.06–0.51)**	**0.001**
Preoperative rupture	No	Reference	
	Yes	**7.1 (1.66–30.2)**	**0.008**
**Surgical route**	Laparoscopy	Reference	
	Laparotomy	0.53 (0.21–1.31)	0.16
	Laparoconversion	0.79 (0.11–5.12)	0.82
Peroperative Rupture	No	Reference	
	Yes	1.09 (0.47–2.49)	0.83
**Type of surgery**			
Cystectomy and contralateral oophorectomy		Reference	
Bilateral cystectomy		0.67 (0.17–2.66)	0.57
Unilateral cystectomy		0.18 (0.05–0.59)	**0.004**
Unilateral oophorectomy		0.23 (0.09–0.57)	**0.001**
**Histology**	Serous	0.82 (0.10–6.15)	0.84
	Mucinous	0.39 (0.05–3.09)	0.37
**FIGO stage**	IA	Reference	
	IB	3.87 (1.10–13.6)	0.03
	IC	2.45 (0.87–6.88)	0.09
	II	**10.2 (2.22–47.3)**	**0.002**
	III	**4.57 (1.82–11.4)**	**0.001**
	other	**3.93 (1.79–8.64)**	**0.0006**
MP Component	No	Reference	
	Yes	**4.35 (1.82–10.3)**	**0.0009**
Implants	No	Reference	
	Yes	**4.95 (2.36–10.4)**	**<0.001**
Microinvasion	No	Reference	
	Yes	1.28 (0.39–4.22)	0.68
		**OR (IC 95%)**	** *p* ** **-value**
**Multivariate analysis**			
Age at diagnosis	≥35 years	Reference	
	≤35 years	**1.40 (1.22–159)**	**0.034**
Nulliparity	No	Reference	
	Yes	**8.04 (1.73–37.4)**	**0.007**
**Type of surgery**			
Cystectomy and contralateral oophorectomy		Reference	
Bilateral cystectomy		1.67 (0.25–10.8)	0.58
Unilateral cystectomy		1.51 (0.22–10.26)	0.67
Unilateral oophorectomy		4.47 (0.13–1.53)	0.20
MP Component	No	Reference	
	Yes	**8.47 (2.42–29.6)**	**0.0008**
Peritoneal Implants	No	Reference	
	Yes	**5.52 (1.8–17.0)**	**0.003**

Data: means (minimum–maximum), and number (%), mm: millimetre, BMI: body mass index, Ca 125: cancer antigen 125, K: cystectomy, USO: unilateral salpingo-oophorectomy, FIGO: International Federation of Obstetrics and Gynaecology, MP: micropapillary.

**Table 6 jcm-11-03645-t006:** (**A**) literature review of oncological results and fertility of patients with advanced stage of serous BOT after conservative surgery. (**B**) literature review of oncological results and fertility of patients with early stages serous BOT after conservative surgery.

**(A)**										
**Authors**	**Year**	**Number of Patients with Stage II/III BOT**	**Median Age at Surgery (years)**	**Median Follow-Up (Months)**	**Number of Patients with Further Pregnancy**	**Median Delay Surgery-Pregnancy, (Months)**	**Number of Recurrences**	**Number of Invasive Recurrence**	**Death**	**Median Time to Recurrence (Months)**
Morice et al.	2001	12	/	/	4 (33.3%)	/	4 (33.3%)	0 (0%)	0 (0%)	/
Camatte et al.	2002	17	25 (14–35)	60 (6–138)	7 (41.2%)	8 (1–55)	9 (52.9%)	2 (11.8%)	0 (0%)	17.5 (5–48)
Uzan et al.	2010	40	26 (14–40)	57 (4–235)	14 (35.9%)	13.5 (3–183)	22 (56.4%)	3 (7.7%)	1 (0.2%)	48 (4–115)
Kane et al.	2010	14	28 (16–40)	36 (16–160)	5 (38.5%)	/	8 (57.1%)	3 (21.4%)	0 (0%)	26 (11–53)
Chanson et al.	2011	5	32.5 (25–34)	71.4(10–135)	4 (80.0%)	/	1 (20.0 %)	2 (40%)	0 (0%)	40 (16–77)
Helpman et al.	2015	59	35	55.3	34 (57.6%)	/	27 (45.8)	/	6 (10%)	30.6
Ziyang Lu et al.	2019	21	28 (22–37)	74 (16–214)	4 (40%)	29 (18–35)	5 (26.3%)	/	0 (0%)	26 (18–53)
Gouy et al.	2020	65	/	/	20 (68.9%)	/	38 (58%)	8 (12.3%)	3 (4.6%)	/
Plett et al.	2020	70	/	/	41 (82.9%)	/	18 (25.5%)	4 (5.7%)	1 (0.3%)	/
**(B)**										
**Authors**	**Year**	**Number of Patients with Stage I BOT**	**Median Age at Surgery (Years)**	**Median Follow-Up (Months)**	**Number of Patients with Further Pregnancy**	**Median Delay Surgery-Pregnancy, (Months)**	**Number of Recurrences**	**Number of Invasive Recurrences**	**Death**	**Median Time to Recurrence (Months)**
Fauvet et al.	2005	162	35.5 (21.9–48.9)	84.8 (32.7–136.9)	21 (38.3%)	28.6 (4–89)	27 (16.6%)	0 (0%)	0 (0%)	39.6
Tinelli et al.	2007	43	28.9	44.5 (4–125)	21 (49%)	28.5 (14–43.2)	3 (7%)	0 (0%)	0 (0%)	30 (28–34)
Yinon et al.	2007	62	28 (13–44)	88	25 (40.3%)	42 (9–144)	16 (25.8%)	/	0 (0%)	36 (7–81)
Park et al.	2009	184	/	/	27 (73%)	/	3(5%)	2 (1%)	1 (0.5%)	65
Uzan et al.	2014	119	29 (11–65)	45 (12–120)	33 (27%)	27	38 (32%)	2 (1.7%)	1 (0.7%)	36.1
Fang et al.	2016	54	28	46.5 (13–146)	36 (68%)	/	19 (35.2%)	0 (0%)	0 (0%)	55
Helpman et al.	2017	112	30 (21.1–38.2)	75	42 (38%)	/	40 (35%)	8 (4.8%)	11 (4.5%)	32
Delle Marchette et al.	2019	535	/	162	252 (47.1%)	/	139 (26%)	/	/	31.5
Chevrot et al.	2020	52	31.9	57	33 (63%)	/	20 (38%)	/	0 (0%)	30.4
Plett et al.	2020	352	33.2	63	41 (82.3%)	/	18 (5.1%)	4 (1.1%)	1 (0.3%)	32 (6–141)

## Data Availability

Data is contained within the article.

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
