# Peer review of "Risk Factors for Recurrence of Borderline Ovarian Tumours after Conservative Surgery and Impact on Fertility: A Multicentre Study by the Francogyn Group"

_jcm, 2022, doi:10.3390/jcm11133645_

Round 1
Reviewer 1 Report
It was a pleasure to review this interesting retrospective multicentre French study assessing clinical factors associated with ovarian borderline tumour recurrence.
The study is quite large and well written and the results are well presented.
The topic doesn't have any novelty on the other side. The study just confirms the extensive literature in the field without adding further new angles of relevant discussion or exploration.
Not clear why in the table they approximate all the values without leaving the decimal. For example, 5.7% becomes 6%.
Percentage of women undergone frozen section and which type of pre-operative radiological imaging between the cohort should be mentioned (if available). If not should be discussed and acknowledged.
I think the discussion is far too long. The authors instead of discussing and reporting the all literature in BOT, which could be read in a review paper, maybe they could focus on what their study adds to this literature and the strength and weakness of their study. At least 2 pages should be cut.
Author Response
It was a pleasure to review this interesting retrospective multicentre French study assessing clinical factors associated with ovarian borderline tumour recurrence.
Thank you
The study is quite large and well written and the results are well presented.
Thank you
The topic doesn't have any novelty on the other side. The study just confirms the extensive literature in the field without adding further new angles of relevant discussion or exploration.
Not clear why in the table they approximate all the values without leaving the decimal. For example, 5.7% becomes 6%.
All the values in the tables were replaced with leaving the decimal
Percentage of women undergone frozen section and which type of pre-operative radiological imaging between the cohort should be mentioned (if available). If not should be discussed and acknowledged.
Data on women who undergone forzen section are not available
161 (92%) women had pelvic ultrasound
91 (52%) had documented pelvic MRI
43 (24.6%) had documented pelvic CT
I think the discussion is far too long. The authors instead of discussing and reporting the all literature in BOT, which could be read in a review paper, maybe they could focus on what their study adds to this literature and the strength and weakness of their study. At least 2 pages should be cut.
As suggested by the reviewer, the discussion was shortened
Reviewer 2 Report
The authors tried to answer an debated question in gyne-oncology field. The recurrences in BOT. The study included patients operated in a time frame of 23 years!!
1)Hovewer, there are only 175 patients. does the sample size is small despite a long period of time?
2) Secondly and importantly, the mean follow up time is only 30 months which is very short during this 23 years of case inclusion. why?
Author Response
1)Hovewer, there are only 175 patients. does the sample size is small despite a long period of time?
The study period is certainly long (23 years) but the incidence of these tumors is rare (639 for the whole population in our centers). The proportion of patients with conservative treatment is even rarer which explains this impression of low size
2) Secondly and importantly, the mean follow up time is only 30 months which is very short during this 23 years of case inclusion. why?
The average follow-up is 30 months. The patients had an instruction of follow-up with the reference center of 5 years but the majority did not return towards the reference center after 2-3 years of follow-up, the patients referred it to their attending physicians and gynecologist
This being said, the patients were always readmitted to the reference center in case of recurrence, except in case of moving
Reviewer 3 Report
This article entitled, “RISK FACTORS FOR RECURRENCE OF BORDERLINE OVARIAN TUMOURS AFTER CONSERVATIVE SURGERY AND IMPACT ON FERTILITY: A MULTICENTRE STUDY BY THE FRANCOGYN GROUP" was one of the largest studies analysing the risk factors for recurrence of BOT after conservative surgery. Although this manuscript includes some contents of great interest to clinicians, the results are not presented appropriately in the text and the English language and style should be changed.
Author Response
This article entitled, “RISK FACTORS FOR RECURRENCE OF BORDERLINE OVARIAN TUMOURS AFTER CONSERVATIVE SURGERY AND IMPACT ON FERTILITY: A MULTICENTRE STUDY BY THE FRANCOGYN GROUP" was one of the largest studies analysing the risk factors for recurrence of BOT after conservative surgery. Although this manuscript includes some contents of great interest to clinicians, the results are not presented appropriately in the text and the English language and style should be changed.
A native English doctor reviewed the English text
the manuscript was modified as requested by the other reviewers
Round 2
Reviewer 1 Report
The authors replied to the comment.
Thanks and congratulations for the manuscript
Author Response
thank you
Reviewer 3 Report
Thank you for giving me this valuable opportunity to review the informative manuscript.
Here are some examples of what I would like to convey to the author.
The authors analysed data from 175 patients with conservative management for BOT in the present study.
・The authors also demonstrated Nulliparity, n (%) is 103(57.5%) in Demographic data. However, 103 divided by 57.5% is not 175.
・The sum of the number of patients who received surgical treatment ( laparoscopy, laparotomy, and conversion) is not 175. There are other similar mistakes. Therefore, I have to say that the quality of the statistical evaluation is not good.
The author should revise their results in this manuscript carefully.
Author Response
we understand the comment of the reviewer, we have added in the tables the number of cases of missing data (NA) that sometimes explains that the sum is not equal to the total n.
In addition, only the percentages that are the result of a manual calculation have all been rechecked, all other statistical results are R generated